# Chloroplast Genome of *Salvia* Sect. *Drymosphace*: Comparative and Phylogenetic Analysis

**Ting Su [1,2], Yan-Fei Geng [3], Chun-Lei Xiang [4], Fei Zhao [4], Mei Wang [1,2], Li Gu [1,2] and Guo-Xiong Hu [1,2,*]**

1 College of Life Sciences, Guizhou University, Guiyang 550025, China; sutingyo@163.com (T.S.); wmgzu20@163.com (M.W.); guligz@163.com (L.G.)
2 The Key Laboratory of Plant Resources Conservation and Germplasm Innovation in Mountainous Region Ministry of Education, Guizhou University, Guiyang 550025, China
3 College of Tea Science, Guizhou University, Guiyang 550025, China; yfgeng@gzu.edu.cn
4 CAS Key Laboratory for Plant Diversity and Biogeography of East Asia, Kunming Institute of Botany, Chinese Academy of Sciences, Kunming 650201, China; xiangchunlei@mail.kib.ac.cn (C.-L.X.); zhaofei@mail.kib.ac.cn (F.Z.)
* Correspondence: gxhu@gzu.edu.cn

**Abstract:** Sect. *Drymosphace* is one of eight sections of *Salvia* subg. *Glutinaria* and includes 13 species and one dubious species that hold great economic value. Although the section is well supported, interspecific relationships remain unresolved. Moreover, most of this section's plastome information remains unknown. In this study, we sequenced and assembled eight sect. *Drymosphace* plastomes and conducted comparative analyses within this section. The length of plastid genome sequences ranged from 151,330 bp to 151,614 bp, with 80 protein-coding, 30 tRNA, and four rRNA genes being annotated. The plastomes were found to be as conservative as other Lamiaceae species, showing high consistency and similarity in terms of gene content, order, and structure. Within the sect. *Drymosphace*, single-copy regions were more variable than IR regions, and the intergenic regions were more variable than the coding regions; nine hypervariable regions were detected, and some of them may be useful for the phylogenetic analysis of *Salvia*. The topologies inferred from all of the data sets indicated that sect. *Drymosphace* was monophyletic and that *S. honania* was sister to *S. meiliensis*. Compared to previous studies involving more sect. *Drymosphace* species, phylogenomic analyses can improve the phylogenetic resolution considerably.

**Keywords:** *Salvia miltiorrhiza*; subg. *Glutinaria*; plastid genome; phylogenomics



## 1. Introduction

The non-monophyletic nature of traditionally defined *Salvia* led to the establishment of a broad definition of *Salvia* that reduces the five small embedded genera (*Rosmarinus*, *Perovskia*, *Dorystaechas*, *Meriandra* and *Zhumeria*) to subgenera, resulting in a total of 11 subgenera being recognized within *Salvia* [1–3]. *Salvia* is the largest genus in Lamiaceae and includes approximately 1000 species. The genus has a subcosmopolitan distribution but is mainly concentrated in South America, Southwest Asia and the Mediterranean region, and East Asia [3]. As one of biodiversity centers of *Salvia*, approximately 100 species have been recorded in East Asia, 85 of which are native to China [4–11]. Based on recent studies, with the exception of *Salvia grandifolia* and *S. deserta* belonging to subg. *Sclarea*, the rest of East Asian *Salvia* have been placed into the newly established subg. *Glutinaria* [2,12].

Sect. *Drymosphace* was first established by George Bentham in 1832–1836 [13] and was later placed in subg. *Salvia* in 1876 [14]. Based on stamen morphology, Stibal [15] transferred sect. *Drymosphace* to subg. *Sclarea*. Following Stibal, Wu [16] also classified sect. *Drymosphace* into subg. *Sclarea* and established three series (ser. *Miltiorrhizae*, ser. *Plectranthoidites*, and ser. *Honaniae*) within this section. According to Wu's circumscription, a total of 19 species and three varieties can be recognized in sect. *Drymosphace* [4,5,9,16],

and all of these taxa are endemic to China, with the exception of *S. nubicola* and *S. plectranthoides*, which extend to some Himalayan countries. Recent phylogenetic studies have demonstrated that sect. *Drymosphace* and the three series sensu Wu [16] are not monophyletic [2,3]. Integrating morphological with molecular phylogenetic evidence, the redefined sect. *Drymosphace* was transferred from subg. *Sclarea* to the newly established subg. *Glutinaria* due to its synapomorphy of robust taproots, pinnate leaves, relatively long corollas (length > 2 cm), and fused deformed posterior thecae [2]. According to the synapomorphy, five species were removed from the previous sect. *Drymosphace* sensu Wu of subg. *Glutinaria* [16]: *S. cavaleriei* and *S. prionitis* were transferred to the newly established sect. *Sobiso*, *S. petrophila* was transferred to sect. *Sonchifoliae*, *S. nubicola* was transferred to sect. *Glutinaria*, *S. trijuga* was transferred to sect. *Substoloniferae*, and *S. breviconnectivata* was regarded as a dubious species [2].

Sect. *Drymosphace* are importance of medical value, with *Salvia miltiorrhiza* being most widely used. "Danshen", a famous traditional Chinese medicine originating from the dried roots of *S. miltiorrhiza*, has been extensively applied to treat coronary heart diseases and cerebrovascular diseases [17]. Due to mass market demand, the wild resource of *S. miltiorrhiza* has reduced sharply. However, although sect. *Drymosphace* is well supported by molecular and morphological evidence, interspecific relationships remain unresolved. Moreover, *S. miltiorrhiza* and its closely related species are morphologically similar, which makes it difficult to identify authentic products, and an increasing number of substitutes and adulterants have been found on the market.

Currently, phylogenetic studies based on short DNA fragments have only been able to determine the phylogenetic backbone of *Salvia* [2,18–22], and relationships among species require further focus. The shallow rate of DNA evolution is one of the main obstacles to determining the relationships between low-level taxa. Due to there being a limited amount of information, the resolutions of phylogenetic trees that has been inferred from chloroplast gene fragments are often low [2,23]. With the birth of next-generation sequencing (NGS) technology, genomics has entered an era of low-cost, large-scale, and high-throughput sequencing, and an increasing number of genomic data are being used for phylogenetic analysis. As one of the three major genomes of green plants, the genomic structure and gene content of chloroplasts is relatively conservative and easy to sequence. At present, plastid genome data are being used to solve phylogenetic relationships at different taxonomic levels and have been demonstrated to be effective [24–26]. Although attempts to utilize NGS to analyze plastome characterization and even though the phylogeny of *Salvia* have been determined, few taxa of sect. *Drymosphace* were involved in these studies [27,28].

In this study, we sequenced eight plastomes of sect. *Drymosphace* and conducted comparative genomics analyses together with two other data sets that have been published on this section (*Salvia meiliensis* and *S. miltiorrhiza*). The aims of this study were (1) to characterize the plastome structure within sect. *Drymosphace*; (2) to screen hyper-variable regions of sect. *Drymosphace*; and (3) to infer the phylogeny of sect. *Drymosphace*.

## 2. Materials and Methods

### 2.1. Plant Materials, DNA Extraction and Genome Sequencing

In this study, 14 taxa (13 species and one variety) were sampled from *Salvia* subg. *Glutinaria*, including 10 taxa from sect. *Drymosphace* (Table 1). Eight sect. *Drymosphace* plastomes were newly sequenced, and the other sequences were downloaded from the GenBank database. Additionally, three species from other subgenera were selected as outgroups (Table 1).

The total genomic DNA was extracted from fresh or silica gel-dried leaves using the modified CTAB method [29]. In order to obtain qualified DNA for library construction, a Qubit Fluorometer was used to determine and calculate the DNA concentration and yield when they were at least c > 12.5 ng/µL and m > 1 µg, and the sample integrity was assessed by means of electrophoresis on a 1% agarose gel. Illumina Hiseq-2500 platform at

BGI-Wuhan was utilized for sequencing by paired-end (PE) library of 150 bp, and 3 to 5 GB of raw short sequence data were generated for each species.

**Table 1.** Voucher information and GenBank and SRA accession numbers for taxa used in this study.

| Taxa | Subgenus | Voucher | Locality | GenBank Accession | SRA Accession |
|---|---|---|---|---|---|
| *S. bowleyana* | *Glutinaria* | GX Hu and F Zhao 131 | China, Fujian | MW435404 * | SRR18650098 * |
| *S. bulleyana* | *Glutinaria* | *NA* | *NA* | MH603954 | *NA* |
| *S. chanryoenica* | *Glutinaria* | s.n. (KH) | South Korea | MH261357 | *NA* |
| *S. dabieshanensis* | *Glutinaria* | GX Hu and F Zhao 0165 | China, Anhui | MW435405 * | SRR18587923 * |
| *S. honania* | *Glutinaria* | GX Hu and F Zhao 0168 | China, Henan | MW435406 * | SRR18600490 * |
| *S. meiliensis* | *Glutinaria* | GX Hu and FZ Shangguan Hu0089 | China, Anhui | MN520018 | *NA* |
| *S. miltiorrhiza* | *Glutinaria* | Cultivated | China, Beijing | HF586694 | *NA* |
| *S. nanchuanensis* | *Glutinaria* | JX Yang and XZ He YJX-01 | China, Hunan | MW435407 * | SRR18595850 * |
| *S. nanchuanensis var. pteridifolia* | *Glutinaria* | GX Hu and B Pan 615 | China, Guangxi | MW435408 * | SRR18595337 * |
| *S. plectranthoides* | *Glutinaria* | GX Hu et al. 0006 | China, Yunnan | MW435409 * | SRR18650097 * |
| *S. prattii* | *Glutinaria* | *NA* | China, Yunnan | MK944407 | *NA* |
| *S. przewalskii* | *Glutinaria* | *NA* | *NA* | MH603953 | *NA* |
| *S. subbipinnata* | *Glutinaria* | JX Yang and XZ He YJX-04 | China, Zhejiang | MW435410 * | SRR18650096 * |
| *S. yunnanensis* | *Glutinaria* | GX Hu 603 | China, Guizhou | MW435411 * | SRR18595336 * |
| *S. hispanica* | *Calosphace* | Cultivated, SD118 | China, Shandong | MT083896 | *NA* |
| *S. rosmarinus* | *Rosmarinus* | Cultivated | China, Shaanxi | KR232566 | *NA* |
| *S. officinalis* | *Salvia* | So_003 | *NA* | MG772529 | *NA* |

*NA* = information is unavailable. * = newly sequenced plastomes in this study. SRA = Sequence Read Archive.

### 2.2. Plastome Assembly and Annotation

After filtering the raw data and discarding the low-quality reads, the remaining PE reads were assembled into whole plastomes using GetOrganelle v1.6.2a [30]. The assembly graph of the generated complete plastome was verified by Bandage v.0.8.1 [31]. Plastome sequences were annotated using software PGA [32], using the *Salvia miltiorrhiza* plastome [33] as reference. The tRNA genes were further verified using the online tRNAscan-SE [34] search servers and then manually adjusted in Geneious 10.0.5 [35]. OrganellarGenomeDRAW, an online program, was used to generate circular annotated plastid genome maps [36], and the plastomes were deposited in the GenBank database (Table 1).

### 2.3. Codon Usage and Repeated Sequence Analysis

Relative Synonymous Codon Usage (RSCU) is a parameter that is used to evaluate the codon usage preferences of protein-coding sequences. Here, the RSCU values for all of the protein-coding sequences were computed using the program codon W 1.4.4 [37]. The codon usage in the form of heatmap was conducted by employing R language with the RSCU value. An RSCU value > 1 indicates that codon usage is highly preferred, an RSCU value = 1 means that codon usage is not preferred, and an RSCU value < 1 indicates that the codon usage is low [38].

The simple sequence repeat (SSR) of sect. *Drymosphace* plastomes were identified using MISA [39] by setting the minimum number of repeat units to 8, 4, 4, 3, 3, and 3 for the mono-, di-, tri-, tetra-, penta-, and hexanucleotides, respectively. Four types of dispersed repeats (forward, reverse, palindrome, and complement repeats) in sect. *Drymosphace* were

determined using REPuter [40] with a repeat size $\geq$ 30 bp and sequence identities $\geq$ 90 (Hamming distance of 3).

### 2.4. Comparative Genomic Analyses

The boundaries of single-copy (SC) and reverse repeat (IR) regions of sect. *Drymosphace* were determined in Geneious 10.0.5, and the boundary expansion and contraction sketch of IR was drawn using Adobe Illustrator CC. mVISTA [41] was used to determine the variability in the complete plastome sequences of sect. *Drymosphace* in Shuffle-LAGAN mode, and *Salvia bowleyana* was selected as the reference genome. Genome rearrangement was carried out by Mauve [42] using *S. bowleyana* as a reference genome. DnaSP v.5.0 [43] was employed to analyze nucleotide diversity (Pi) within sect. *Drymosphace* by setting the step size to 200 bp with a 600 bp window length.

### 2.5. Phylogenetic Analysis

To infer the phylogenetic relationships of sect. *Drymosphace*, 14 plastid genomes from subg. *Glutinaria* were selected to carry out analyses with *S. rosmarinus*, *S. hispanica*, and *S. officinalis* from three other subgenera as outgroups. Seven matrices, including complete plastome (CP) sequences with one IR region excluded, large single copy (LSC), small single copy (SSC), IR, protein-coding exons (coding regions, CR), intergenic spacer and introns (non-coding region, NCR), and nine hyper-variable region (HVR) data sets were prepared for the phylogenetic analyses. The sequences were aligned with MAFFT [44] using Gblocks v0.91 [45] to exclude the ambiguously aligned positions. The neat sequence matrices were employed to infer phylogenetic trees using both maximum likelihood (ML) and Bayesian inference (BI).

Under the GTRGAMMA substitution model, ML analyses were performed using RAxML-HPC2 on XSEDE v.8.2.12 [46] on the CIPRES Science Gateway (http://www.phylo. org/ (accessed on 20 March 2022)) [47]. With the exception of setting bootstrap iterations (–# | –N) to 1000, the other parameters used default values.

BI analysis was carried out in MrBayes 3.2.6 [48] and implemented in PhyloSuite [49]. The ModelFinder [50] was utilized to select the best model according to the Akaike information criterion (AIC) (Table 2). Four Markov chain Monte Carlo (MCMC) iterations were run simultaneously for 2,000,000 generations. Each run started with a random tree, and a random tree was sampled every 1000 generations. Stationarity was considered to be reached when the average standard deviation of the split frequencies was less than 0.01. After discarding the first 25% trees as burn-in, the remaining trees were used to calculate a majority-rule consensus tree for each matrix.

**Table 2.** Summary of the data set information for phylogenetic analyses.

| Data Matrix | Aligned Length (bp) | Constant Sites (bp) | Variable Sites (bp) | Parsimony Informative (bp) | Best Fit Model (AIC) |
|:---:|:---:|:---:|:---:|:---:|:---:|
| CP | 151,218 | 144,550 (95.59%) | 6668 | 2366 (1.56%) | GTR + F + I + G4 |
| LSC | 82,556 | 77,911 (94.37%) | 4645 | 1637 (1.98%) | GTR + F + I + G4 |
| SSC | 17,531 | 15,967 (91.08%) | 1564 | 567 (3.23%) | GTR + F + G4 |
| IR | 25,563 | 25,335 (99.11%) | 228 | 80 (0.31%) | GTR + F + I |
| CR | 66,888 | 64,304 (96.14%) | 2584 | 921 (1.38%) | GTR + F + I + G4 |
| NCR | 59,877 | 56,165 (93.80%) | 3712 | 1346 (2.25%) | GTR + F + I + G4 |
| HVR | 10,298 | 9096 (88.33%) | 1064 | 363 (3.52%) | GTR + F + G4 |

## 3. Results and Discussion

### 3.1. Plastome Features of Sect. Drymosphace

The length of complete plastomes of sect. *Drymosphace* ranged from 151,330 bp (*Salvia dabieshanensis*) to 151,614 bp (*S. meiliensis*). All of the plastomes displayed a typical quadripartite architecture that contained a large single copy (LSC: 82,629–82,854 bp) and a small

single copy (SSC: 17,541–17,587 bp) separated by two copies of an inverted repeat (IR: 25,520–25,590 bp). The total GC content varied slightly from 38.00% to 38.03%, and the IR regions had the highest GC content (43.10–43.14%) followed by those in the LSC (36.12–36.15%) and SSC (31.97–32.05%) regions (Figure 1, Table 3).

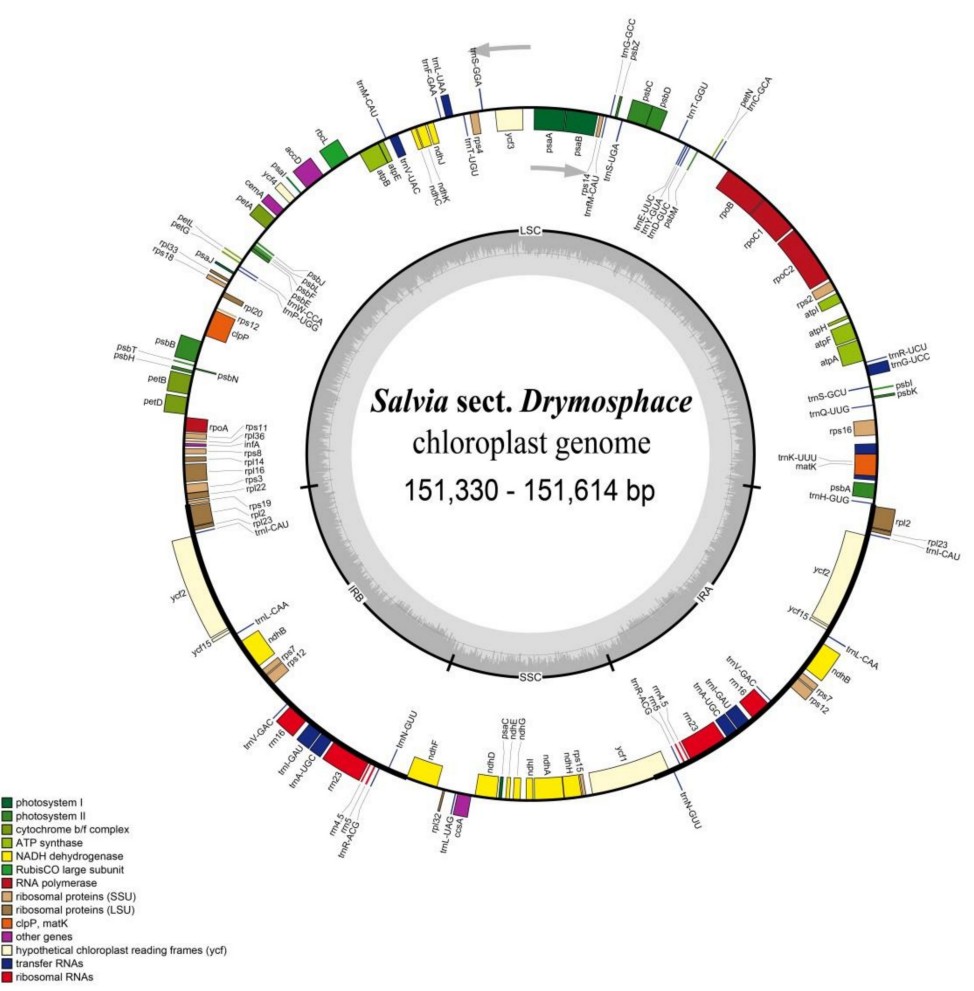

**Figure 1.** Complete plastome gene map of *Salvia* sect. *Drymosphace*. Genes outside of the circle are transcribed in the counterclockwise direction, and those that are inside are transcribed in the clockwise direction. LSC—large single copy; SSC—small single copy; IR—inverted repeat.

A total of 114 unique genes, including 80 protein-coding, 30 tRNA, and four rRNA genes, were detected in each species (Table 3). Four rRNA, seven tRNA, and seven protein-coding genes were duplicated in IR regions (Table 3). Of the 18 genes containing intron, 15 comprised only one intron, and three genes harbored two introns. The *rps12* gene was recognized as being a trans-spliced gene, with the 5′ end located in the LSC region and the duplicated 3′ end in the IR region (Figure 1, Table 4).

**Table 3.** Plastome features of 10 taxa of *Salvia* sect. *Drymosphace* presented in this study.

| Taxa | Clean Reads | Reads Used for Assembly | Mean Coverage (×) | Complete | | LSC | | SSC | | IR | | Number of Genes | PCG | tRNA Genes | rRNA Genes |
|---|---|---|---|---|---|---|---|---|---|---|---|---|---|---|---|
| | | | | Size(bp) | GC (%) | Length (bp) | GC (%) | Length (bp) | GC (%) | Length (bp) | GC (%) | | | | |
| *S. bowleyana* | 6961,322 | 6,160,014 | 526 | 151,508 | 38.01 | 82,809 | 36.14 | 17,585 | 32.01 | 25,557 | 43.12 | 114 | 80 | 30 | 4 |
| *S. dabieshanensis* | 11,727,732 | 9,644,622 | 504 | 151,330 | 38.02 | 82,629 | 36.14 | 17,587 | 32.01 | 25,557 | 43.12 | 114 | 80 | 30 | 4 |
| *S. honania* | 7,221,162 | 6,369,236 | 514 | 151,522 | 38.01 | 82,768 | 36.13 | 17,586 | 31.97 | 25,584 | 43.11 | 114 | 80 | 30 | 4 |
| *S. meiliensis* | 68,112,046 | 25,577,470 | 1970 | 151,614 | 38.00 | 82,854 | 36.12 | 17,580 | 32.00 | 25,590 | 43.10 | 114 | 80 | 30 | 4 |
| *S. miltiorrhiza* | NA | NA | NA | 151,332 | 38.02 | 82,698 | 36.15 | 17,556 | 32.01 | 25,539 | 43.12 | 114 | 80 | 30 | 4 |
| *S. nanchuanensis* | 9,188,568 | 7,579,704 | 507 | 151,426 | 38.02 | 82,791 | 36.14 | 17,563 | 31.99 | 25,536 | 43.13 | 114 | 80 | 30 | 4 |
| *S. nanchuanensis* var. *pteridifolia* | 15,948,806 | 13,152,169 | 514 | 151,455 | 38.01 | 82,831 | 36.12 | 17,584 | 32.05 | 25,520 | 43.14 | 114 | 80 | 30 | 4 |
| *S. plectranthoides* | 6,978,570 | 6,231,469 | 512 | 151,416 | 38.01 | 82,775 | 36.13 | 17,563 | 32.03 | 25,539 | 43.12 | 114 | 80 | 30 | 4 |
| *S. subbipinnata* | 5,368,560 | 4,649,062 | 516 | 151,388 | 38.03 | 82,769 | 36.15 | 17,541 | 32.02 | 25,539 | 43.13 | 114 | 80 | 30 | 4 |
| *S. yunnanensis* | 6,448,098 | 5,394,701 | 508 | 151,413 | 38.02 | 82,652 | 36.14 | 17,581 | 32.03 | 25,590 | 43.11 | 114 | 80 | 30 | 4 |

**Table 4.** Genes present in the plastomes of 10 taxa of sect. *Drymosphace*.

| Gene Functions | Group of Genes | Name of Genes |
|---|---|---|
| Photosynthesis | Subunits of ATP synthase | *atpA, atpB, atpE, atpF *, atpH, atpI* |
| | Subunits of NADH dehydrogenase | *ndhA *, ndhB * (×2), ndhC, ndhD, ndhE, ndhF, ndhG, ndhH, ndhI, ndhJ, ndhK* |
| | Subunits of cytochrome | *petA, petB *, petD *, petG, petL, petN* |
| | Subunits of photosystem I | *psaA, psaB, psaC, psaI, psaJ* |
| | Subunits of photosystem II | *psbA, psbB, psbC, psbD, psbE, psbF, psbH, psbI, psbJ, psbK, psbL, psbM, psbN, psbT, psbZ* |
| | Subunit of rubisco | *rbcL* |
| Self-replication | Large subunit of ribosome | *rpl2 * (×2), rpl14, rpl16 *, rpl20, rpl22, rpl23 (×2), rpl32, rpl33, rpl36* |
| | DNA-dependent RNA polymerase | *rpoA, rpoB, rpoC1 *, rpoC2* |
| | Small subunit of ribosome | *rps2, rps3, rps4, rps7 (×2), rps8, rps11, rps12 ** (×2), rps14, rps15, rps16*, rps18, rps19* |
| | rRNA Genes | *rrn4.5 (×2), rrn5 (×2), rrn16 (×2), rrn23 (×2)* |
| | tRNA Genes | *trnA-UGC * (×2), trnC-GCA, trnD-GUC, trnE-UUC, trnF-GAA, trnfM-CAU, trnG-GCC, trnG-UCC *, trnH-GUG, trnI-CAU (×2), trnI-GAU * (×2), trnK-UUU *, trnL-CAA (×2), trnL-UAA *, trnL-UAG, trnM-CAU, trnN-GUU (×2), trnP-UGG, trnQ-UUG, trnR-ACG (×2), trnR-UCU, trnS-GCU, trnS-GGA, trnS-UGA, trnT-GGU, trnT-UGU, trnV-GAC (×2), trnV-UAC *, trnW-CCA, trnY-GUA* |
| Other genes | Subunit of Acetyl-CoA-carboxylase | *accD* |
| | c-type cytochrome synthesis gene | *ccsA* |
| | Envelop membrane protein | *cemA* |
| | Protease | *clpP ** |
| | Translational initiation | *infA* |
| | Maturase | *matK* |
| Unknown function | Conserved open reading frames | *ycf1, ycf2 (×2), ycf3 **, ycf4, ycf15 (×2)* |

* gene with a single intron; ** gene with two introns; (×2) duplicated gene.

The plastid genomes of sect. *Drymosphace* showed high consistency and similarity in terms of their gene structure, order, and content. These results are consistent with other plastomes of Lamiaceae [26–28,33,51] and are also as conservative as most of the angiosperms that have been previously reported [52–56].

### 3.2. Amino Acid Abundance and Codon Usage

The number of codons in sect. *Drymosphace* ranged from 26,501 (*Salvia miltiorrhiza*) to 26,583 (*S. dabieshanensis*). Among these codons, leucine (10.59–10.64%) was the most frequently observed amino acid, and cysteine (1.10–1.12%) was the least frequently observed (Figure 2, Table S1). Most protein-coding genes had the same standard ATG sequence as the initiation codon, but the *rps19* and *ycf15* genes started with a GTG sequence. This non-ATG initiation codon has also been found in other angiosperm chloroplasts [28,57,58]. Heatmaps showed that 30 of the total 64 types of codons used in sect. *Drymosphace* were preferred codons (RSCU > 1), and with the exception of UUG (Leu), all of the preferred codons ended with an A or a U (Figure 3, Supplementary Material Table S1). This codon usage bias shows a similar trend observed in the majority of plastomes in higher plants [52,59,60].

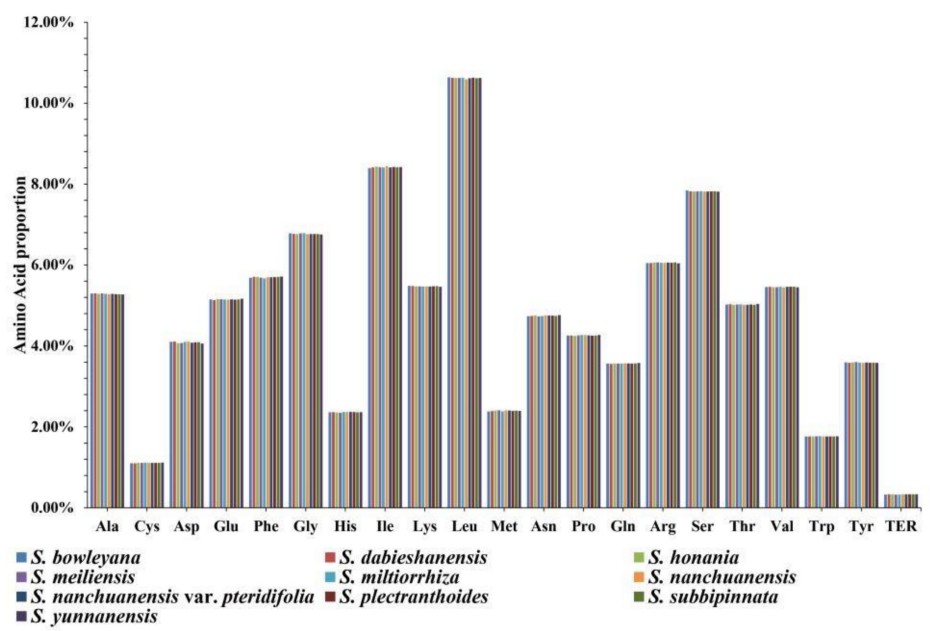

**Figure 2.** Amino acid frequencies in all protein-coding genes in the chloroplast genome of 10 taxa of *Salvia* sect. *Drymosphace*.

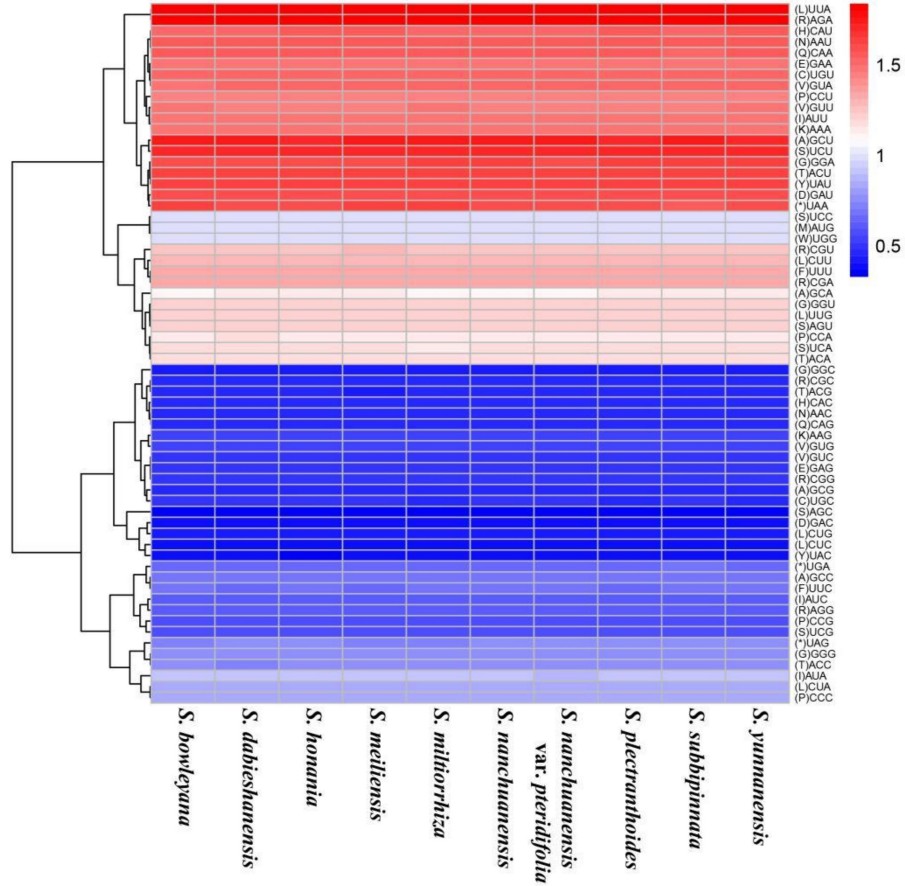

**Figure 3.** Heatmap of codon distribution of all protein-coding genes from 10 taxa of *Salvia* sect. *Drymosphace*. Higher red values indicate higher RSCU values, and lower blue values indicate lower RSCU values.

### 3.3. Simple Sequence Repeats and Long Repeat Sequences

Simple sequence repeat (SSR), also called microsatellite, is a sequence with a tandemly repeated motif that ranges in size from one to six. Due to high repeatability and polymorphism, SSR is a molecular marker that is commonly used in population genetics and evolutionary analysis [52,61–64]. In this study, a total of 1685 SSRs were identified with each taxon bearing 166–173 SSRs from 10 plastomes of sect. *Drymosphace* (Figure 4, Table S2). Five types of SSR (excluding penta-nucleotide SSR) were detected in sect. *Drymosphace*, of which the mono-nucleotide SSR was the richest (1265, 75.07%), followed by the di-nucleotide (346, 20.53%), tetra-nucleotide (69, 4.09%), hexa-nucleotide (4, 0.24%), and tri-nucleotide (1, 0.06%) SSRs. In terms of repeat units, the A/T unit in the mono-nucleotide repeat was the most abundant (1242, 73.71%), followed by the di-nucleotide AT/AT (136, 8.07%) and TA/TA (100, 5.93%). The tri-nucleotide SSR (AAT/ATT) was only detected once (1, 0.06%) in *S. nanchuanensis* var. *pteridifolia*, and the hexa-nucleotide SSRs (4, 0.24%) were detected once in *S. dabieshanensis* (ATTCAT/ATGAAT), *S. meiliensis* (AATCAA/TTGATT), *S. miltiorrhiza* (ACTTAG/CTAAGT), and *S. plectranthoides* (ACTTAG/CTAAGT). In addition, six unique repeat units (AAT/ATT, AAAT/ATTT, TTAA/AATT, ATTCAT/ATGAAT, AATCAA/TTGATT, ACTTAG/CTAAGT) were detected within sect. *Drymosphace*. The majority of the SSRs resided in LSC regions (1125, 66.77%), and the others were located in other SSC regions (320, 18.99%) and in IR regions (240, 14.24%). As reported in previous plastomes of *Salvia* of Lamiaceae [28,51] as well as in other families [55,65,66], most of the SSRs in sect. *Drymosphace* are composed of A/T mono-nucleotides, which represent a percentage of 73.71%. Similar to most *Salvia* plastomes that have been reported upon before [28,51,67], we did not detect the penta-nucleotide SSR in sect. *Drymosphace*. However, a recent study reported a penta-nucleotide SSR from *S. yunnanensis* of sect. *Drymosphace* [68]. *Salvia yunnanensis* was also included in this study as well as in another independent analysis [28], in which the samples were from Guizhou and Yunnan Province, China, respectively, but neither study identified the penta-nucleotide SSR in these species. Therefore, there is a lack of the penta-nucleotide within sect. *Drymosphace*, or at the very least, it is rare. Even though the mono-nucleotide SSRs showed significant differences among species, ranging from 123 (*S. miltiorrhiza*) to 134 (*S. yunnanensis*), no significant differences were observed in the number of the other four types of SSRs.

A total of 478 long repeat sequences with lengths greater than 30 bp, including 222 forward and 255 palindromic repeats and one complementary repeat, were detected in sect. *Drymosphace* (Figure 5, Table S3). The number of repeats for each taxa varied from 42 to 49 and had lengths that ranged from 30 to 99 bp. Most of the long repeats were located in IR (389, 81.38%) regions, followed by LSC (79, 16.53%) and SSC (10, 2.09%). In particular, most of these repeats were detected in coding regions (378, 79.08%), with a few being located in intergenic regions (50, 10.46%) and in introns (50, 10.46%). Long repeat sequences may play a pivotal t role in plant evolution and could promote variation and rearrangement in the plastid genome [69–71]. All of these repeats, together with the aforementioned SSRs, may have potential utility in population studies.

### 3.4. Comparative Genomic Analyses

The contraction and expansion of IR regions at the borders is a main reason for size variation in plastomes and plays an important role in the evolution of seed plants [72–75]. To explore the expansion and contraction of the IR regions, the boundaries between the SC and IR regions of sect. *Drymosphace* were analyzed. The *rps19* gene spanned the LSC/IRb boundary in all species and had a length of 234 bp or 236 bp in the LSC and 43 bp or 45 bp in the IRb, which generated a short *rps19* pseudogene (*ψrps19*) that was 43 bp or 45 bp in length at the IRa/LSC border. The *ndhF* crossed the IRb/SSC border and had an equal length of 32 bp in the IRb and of 2185 bp or 2203 bp in the SSC. The SSC/IRa boundary was located within the *ycf1* gene and had lengths ranging from 4452 bp to 4470 bp in the SSC and of 1056 bp or 1167 bp in the IRa, resulting in a pseudogene (*ψycf1*) at the IRb/SSC border with 1056 bp or 1167 bp overlapping with the *ndhF* gene. In addition, the IRa/LSC

boundary resided between the *ψrps19* and *trnH*, and the distance from *trnH* was 8 bp or 10 bp (Figure 6). In *Salvia*, IR/LSC junctions are considered to be highly conserved, and the expansion/contraction of the IR regions is infrequent [28]. Although a few IR contractions have been reported in *Salvia* (e.g., *S. hispanica*, *S. mekongensis*, and *S. rosmarinus*) [28], no contractions were detected in this study, suggesting that IR contraction events may not occur in sect. *Drymosphace*.

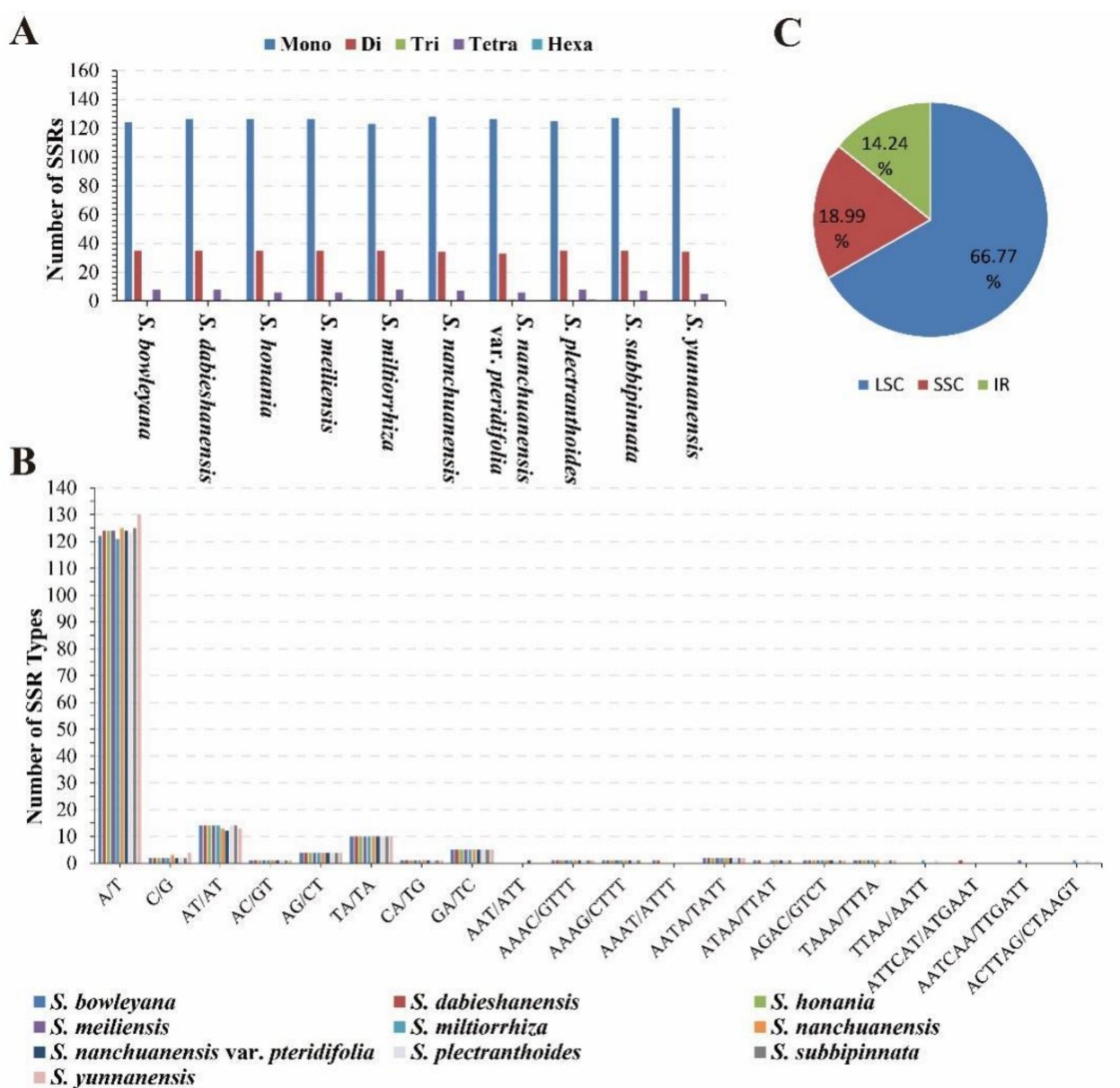

**Figure 4.** Comparison of simple sequence repeats (SSR) among the plastomes of 10 taxa of *Salvia* sect. *Drymosphace*. (**A**). Number of different types of SSRs; (**B**). number of different of SSR repeat units; (**C**) frequencies of SSRs in LSC, IR, and SSC regions.

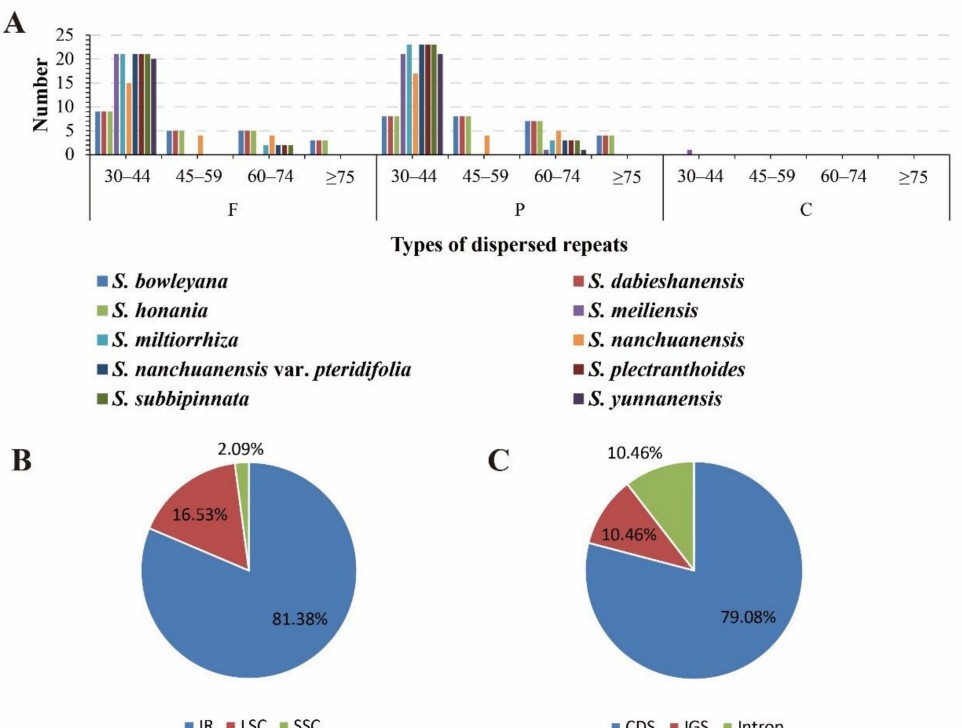

**Figure 5.** Analysis of repetitive sequences in the plastomes of 10 taxa of *Salvia* sect. *Drymosphace*. (**A**). Number of different types of longer repeats; (**B**). frequencies of longer repeats in the LSC, SSC, and IR regions; (**C**). frequencies of longer repeats in protein coding regions, intergenic spacers, and intron regions.

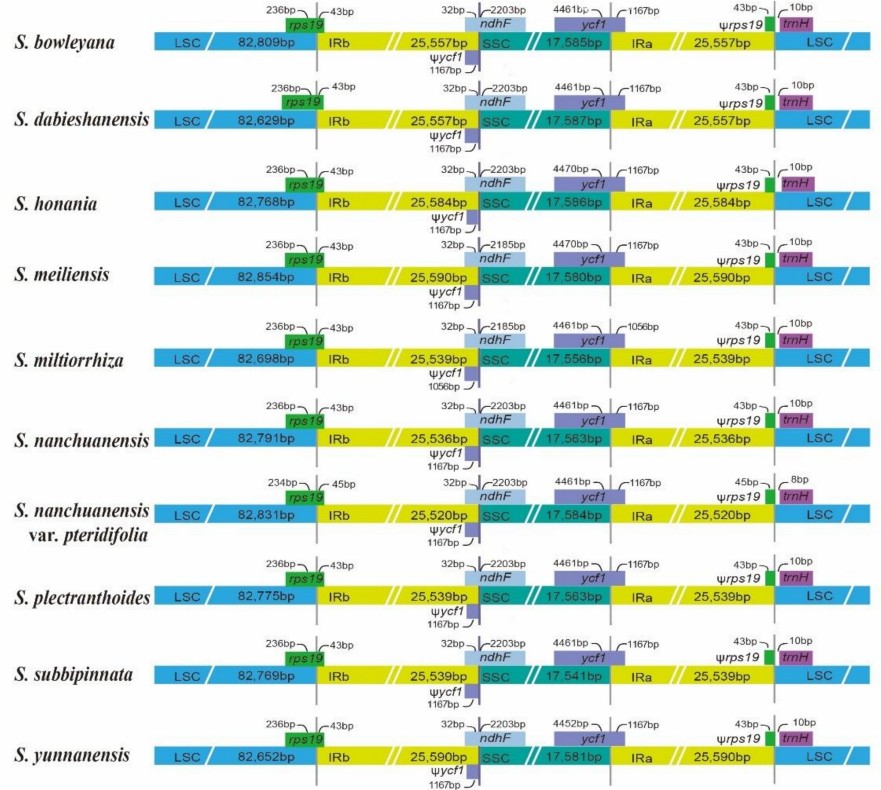

**Figure 6.** Comparison of junctions between the LSC, IR, and SSC regions among plastomes of 10 taxa of *Salvia* sect. *Drymosphace*. Genes are denoted by colored boxes, with *ψ* indicating a pseudogene.

Comparison of complete plastome sequences showed that the genes within sect. *Drymosphace* were arranged in the same order and that the IR regions were more conserved than the LSC and SSC regions (Figure 7), which may be related to the copy corrections caused by gene conversion between the two IR regions [76,77]. In addition, noncoding regions were found to be more divergent than coding regions, and the most varied regions occurred in intergenic regions, such as *accD-psaI*, *rps4-trnL*, *rps16-trnQ*, and *trnH-psbA*. Genome rearrangement analysis showed a collinear relationship, suggesting that there was no rearrangement or inversion events in the plastomes of sect. *Drymosphace* (Figure S1).

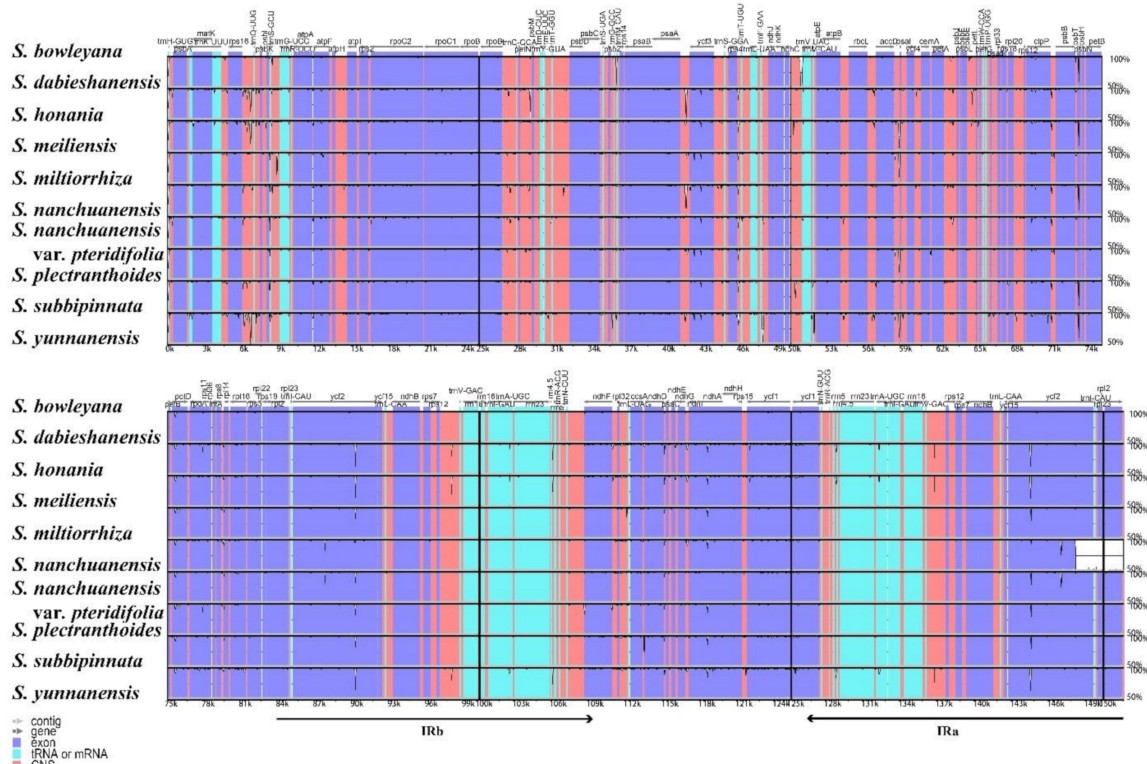

**Figure 7.** Sequence alignment of the whole plastomes of 10 taxa of *Salvia* sect. *Drymosphace* using mVISTA with *S. bowleyana* as a reference. The vertical scale indicates percentage of identity, ranging from 50% to 100%.

Sequence divergence analyses showed that the nucleotide variability (Pi) values of the sect. *Drymosphace* plastomes ranged from 0 to 0.01376, with an average value of 0.0014 (Figure 8). Nine highly variable regions with Pi values > 0.005 were detected, including two genes (*clpP* and *ycf1*) and seven intergenic spacers (*trnH-psbA*, *trnK-rps16*, *petN-psbM*, *rps4-trnT*, *rbcL-accD*, *rpl32-trnL-ccsA*, and *ndhD-psaC*). Among these hyper-variable regions, six loci were located in the LSC, three were located in the SSC, and none were located in the IR regions. *trnH-psbA* is the most variable region (Pi = 0.01376) among the nine hypervariable regions. Due to high rates of insertion/deletion and universal primers, *trnH-psbA* has been selected to be a plant barcode for species discrimination [78–81] and has been used for phylogenetic analyses in Lamiaceae [82–84]. However, the resolution of this DNA marker is not high enough to solve phylogenetic relationships at the section and below levels for *Salvia* [2,85]. For the two hypervariable genic regions, the *clpP* gene has not been used as a marker in phylogenetic study of *Salvia*, and *ycf1* has been demonstrated to be a promising maker for phylogenetic analyses within *Salvia* [28,86].

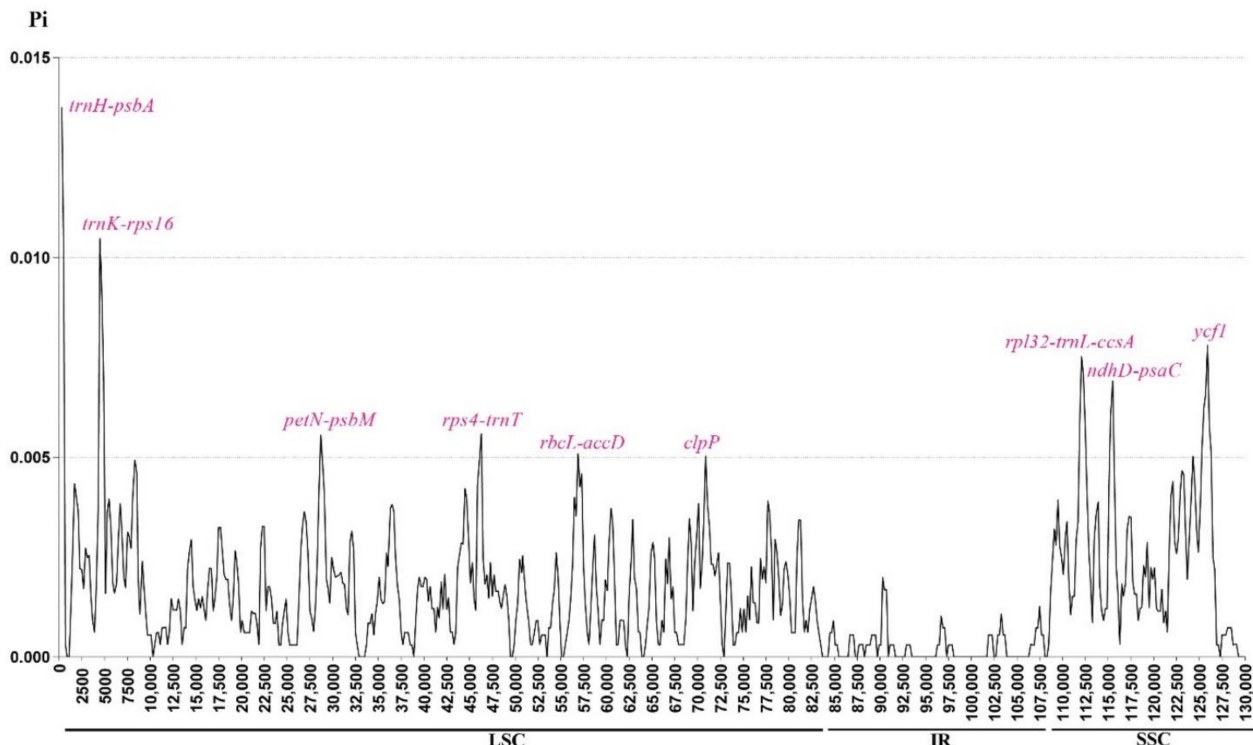

**Figure 8.** Sliding window analysis of plastomes of 10 taxa of *Salvia* sect. *Drymosphace*. The *X*-axis indicates the position of the midpoint of a window, and the *Y*-axis represents the nucleotide diversity of each window. The nine hyper-variable regions with Pi > 0.005 are marked.

*3.5. Phylogenetic Analysis*

Seven matrices including 17 taxa were employed to conduct phylogenetic analyses (Table 1). The aligned CP dataset had the longest length (145,835 bp), and the combined nine HVRs had the shortest length (10,298 bp). The highest percentage of constant sites were from the IR data set (99.11%), and the lowest were from the HVR data set (88.33%). As opposed to the constant sites, the percentage of the informative sites in the HVR data set was the highest (3.52%), and lowest percentage was in the IR regions (0.31%) (Table 2).

The topologies that were inferred by ML and BI analyses for each matrix were identical, with the exception of a few weakly supported nodes. Therefore, only an ML tree with posterior probabilities (PP) indicated above the branches and the ML bootstrap (BS) values provided below.

In all of the analyses, monophylies of subg. *Glutinaria* and sect. *Drymosphace* were recovered, and sister relationships between *Salvia honania* and *S. meiliensis*, *S. dabiesha-nensis*, and *S. bowleyana* within sect. *Drymosphace* were strongly supported (Figure 9 and Figures S2–S7). Within sect. *Drymosphace*, with the exception of the CR matrix, all of the analyses supported the finding that *S. honania* and *S. meiliensis* together to be sister to the rest of the section; with the exception of the IR regions, all of the topologies that were inferred from other data set showed that *S. dabieshanensis*, *S. bowleyana*, *S. miltiorrhiza*, and *S. plectranthoides* formed a strongly supported subclade. The phylogenetic positions of *S. nanchuanensis*, *S. yunnanensis*, and *S. subbipinnata* varied slightly in different data set.

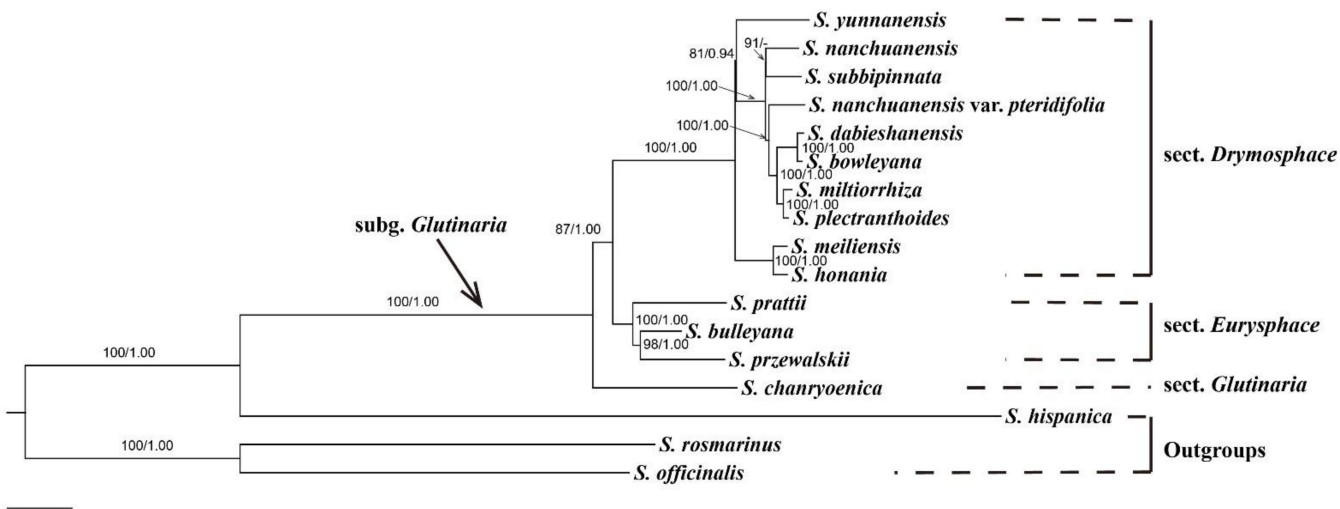

**Figure 9.** Phylogenetic tree inferred from maximum likelihood and Bayesian inference based on complete chloroplast genome. Posterior probabilities are provided above the branches, and the ML bootstrap values are indicated below, with the ML bootstrap values <50% and PP < 0.90 indicated by '-'.

Integrating molecular phylogenetic with morphological evidence, Hu et al. [2] established a new sect. *Drymosphace*, but the interspecific relationships within the section are confused, and the supported values for most subclades are weak. For example, *S. honania* and *S. meiliensis* are two morphologically similar species [87]. They are easily distinguished from the other species within sect. *Drymosphace*, as characterized by small and oblong upper corolla lips and clearly exserted stamens that adhere laterally to the corolla [2,87]. The two species should have been sister group. However, they did not gather together in previous analyses that were based on only a few DNA markers [87]. In this study, all of the analyses indicated that the two species formed a sister relationship, supporting the finding that *S. honania* is closely related to *S. meiliensis* [2,87]. Additionally, compared to previous studies including sect. *Drymosphace* based on short DNA fragments [2,12,85], the plastid phylogenomic analyses in this study based on all of the matrices greatly improved the phylogenetic resolution (Figure 9). Although monophyly of sect. *Drymosphace* and sister relationships of *S. honania* and *S. meiliensis* were confirmed in this study, the relationships between some of the taxa presented here are inconsistent with those inferred by morphological characteristics. Morphologically, *Salvia plectranthoides*, *S. nanchuanensis*, and *S. nanchuanensis* var. *pteridifolia* share long tubular corolla tubes and straight upper corolla lips and are therefore placed in ser. *Plectranthoidites* [4,16]. They should have been found to be closely related and grouped together in phylogenetic trees, especially *S. nanchuanensis* and *S. nanchuanensis* var. *pteridifolia*. However, in this study, *S. nanchuanensis* is sister to *S. subbipinnata* instead of *S. nanchuanensis* var. *pteridifolia*, and *S. plectranthoides* is sister to *S. miltiorrhiza*, indicating that *S. nanchuanensis* and ser. *Plectranthoidites* are not monophyletic. Therefore, the plastid genome is still unable to completely determine the interspecific relationships of sect. *Drymosphace*.

Of all seven data sets, the IR regions generated the poorest phylogenetic resolution, with four subclades having very low support values within sect. *Drymosphace* (PP < 0.9 and BS < 50%) (Figure S4). Wu et al. [27] also reported similar results in plastid phylogenomic results for *Salvia* [27]. The sequence length as well as informative sites may lead to this result. Previous studies indicate that more and longer DNA sequences may greatly improve the phylogenetic resolution and the support values of the branches [28,88,89]. In this study, the IR regions (25,563 bp) and HVR (10,298) were the two shortest data sets, so the phylogenetic resolutions that were inferred from them are lower than those of the other five matrices (Figures S4 and S7). On the other hand, the IR region has the lowest percentage of

informative sites, which may be the result of the low resolution. Therefore, the use of IR regions separately is not suggested for phylogenetic analysis in *Salvia*.

## 4. Conclusions

In this study, we sequenced and assembled eight *Salvia* sect. *Drymosphace* plastomes. The reported plastid genomes are conservative, showing high levels of consistency and similarity to the chloroplast genomes of other species of Lamiaceae in terms of gene content, order, and structure. Comparative analyses revealed that there is no rearrangement or inversion events in the plastomes of *sect. Drymosphace* and that single-copy regions and intergenic regions are more variable than IR regions and coding regions. Phylogenomic analyses can recover the monophyly of the newly established sect. *Drymosphace*, considerably improve phylogenetic resolution, and determine interspecific relationships for some of the species within this section.

**Supplementary Materials:** The following are available at https://www.mdpi.com/article/10.3390/d14050324/s1. Table S1: Amino acid frequencies and codon distribution of 10 taxa of *Salvia* sect. *Drymosphace*, Table S2: SSRs of 10 taxa of *Salvia* sect. *Drymosphace*. Table S3: Long repeats of 10 taxa of *Salvia* sect. *Drymosphace*, Figure S1: Mauve alignment of plastomes of 10 taxa of *Salvia* sect. *Drymosphace*, Figure S2: Phylogenetic tree inferred from maximum likelihood and Bayesian inference based on LSC, Figure S3: Phylogenetic tree inferred from maximum likelihood and Bayesian inference based on SSC, Figure S4: Phylogenetic tree inferred from maximum likelihood and Bayesian inference based on IR, Figure S5: Phylogenetic tree inferred from maximum likelihood and Bayesian inference based on CR, Figure S6: Phylogenetic tree inferred from maximum likelihood and Bayesian inference based on NCR, Figure S7: Phylogenetic tree inferred from maximum likelihood and Bayesian inference based on HVR.

**Author Contributions:** Conceptualization, G.-X.H.; software, T.S., L.G. and F.Z.; formal analysis, T.S., M.W. and L.G.; writing—original draft preparation, T.S. and G.-X.H.; writing—review and editing, G.-X.H., C.-L.X., T.S., Y.-F.G. and F.Z.; funding acquisition, G.-X.H. and Y.-F.G. All authors have read and agreed to the published version of the manuscript.

**Funding:** This research was funded by the National Natural Science Foundation of China (32060048 and 31900275) and the Construction Program of Biology First-Class Discipline in Guizhou (GNYL [2017] 009).

**Institutional Review Board Statement:** Not applicable.

**Informed Consent Statement:** Not applicable.

**Data Availability Statement:** The plastome sequence data supporting this study are openly available in the GenBank nucleotide database and the Sequence Read Archive. Other data are contained within the article or within the Supplementary Materials.

**Acknowledgments:** We thank Jia-Xin Yang, Xuanze He, and Bo Pan for their help with the field work.

**Conflicts of Interest:** The authors declare no conflict of interest.

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
