# Peer review of "Chloroplast Genome of Salvia Sect. Drymosphace: Comparative and Phylogenetic Analysis"

_diversity, doi:10.3390/d14050324_

Round 1
Reviewer 1 Report
The reviewed manuscript by Su et al. provides some novel genetic resources of Salvia genus by sequencing and assembling eight plastosomes of Drymosphace section. Authors in detail described newly assembled plastomes and provide basic comparative statistics. I strongly believe that obtained data have some potential, but additional analyses are required to expand the descriptive character of the manuscript. Also, due to several methodological flaws I can't recommend it for publication in Diversity in the present form.
Major issues
Generally, the manuscript lacks discussion. There are just a few sentences at the end of each subtitle. This part should be most important and the current form is unacceptable. I would recommend separation results from discussion in this particular case.
The matrices for phylogenetics partitions shouldn’t be selecter a priori. The whole alignment should be partitioned according to evolutionary models using software like partitionfinder, and then one final tree should be calculated using MrBayes or RAxML (or both). It can be easily done using pipelines like Phylosuite. Subtrees can be calculated using f.e. CDS and IGS/intronic dataset. Moreover, the manuscript would gain from wider selection of Salvia plastomes - there’s over 100 present in GenBank
Codon usage analysis should be evaluated using modern statistical software using more advanced models like MILC. There are no details about therehold used during this work and CU analysis confidence suffers from short, usually below 80 AA long sequences.
Conclusions should be rewritten too, since half of this paragraph has no conclusive character but rather summarizing the results.
Next improvement could be made on molecular species identification, which is not mentioned at all. The plastid superbarcodes are gaining popularity and sequenced plastomes should be analyzed using one the modern species delimitation approach (DN, MDCs or one of the tree based methods).
Minor issues:
Lines 266-270: only 3 out 7 references are about barcoding. Comparative analysis is not a barcoding approach. And trnH-psbA is unefficient barcode for Lamiaceae (Krawczyk et al. 2014).
Tables 1 and 2 should be merged removing non-variable columns from Table 2. Additionally the coverage statistics should be given within this table. I would also recommend providing SRA accession for sequenced reads.
An analysis presented in figure 5 should be normalized by the total length of each part. It would be nice to visualize distribution of repeats in a consensus genome map and check the dispersion of repeats at phylogenetic trees.
Figure 6 should be removed, since it is uninformative - all plastomes are identical, without any changes at boundaries.
Figure 7 - please spit it in at least four parts, since it’s unreadable in present form
Figure 8 - as the figure 6, it brings nothing to the table.
Reviewer 2 Report
I am sending my comments in the attached file.

Reviewer 3 Report
Dear author,
I suggest you add this reference into your article in introduction or discussion.
https://onlinelibrary.wiley.com/doi/10.1111/j.1759-6831.2012.00232.x
Phylogenetic relationships of Salvia (Lamiaceae) in China: Evidence from DNA sequence datasets by Qian-Quan LI,Min-Hui LI,Qing-Jun YUAN,Zhan-Hu CUI,Lu-Qi HUANG,Pei-Gen XIAO.
The methods have no problems.
The results are incorporated into the research questions and experimental designs.
Basically, the introduction writes down the proper information.
Finally, although this study uses new methods to solve the systematic issues, there are no big differences with other traditional methods or other methods.
You may want to discuss more in systematics and evolution to make this manuscript more interesting.
Reviewer 4 Report
General comments:
1. Please improve the language usage, including listed below and unlisted.
2. Please provide one major phylogram (with branch lengths) rather than seven cladograms. These cladograms can be placed as smaller subgraphs or moved into the supplement.
Details:
Line 21 & 315: consistence -> consistency; please correct it throughout the entire manuscript; https://english.stackexchange.com/questions/40564/consistence-and-consistency
Line 25: Comparing -> Compared
Line 58 & 71: rewrite the sentences concerning "as", "which". Avoid using subordinate clauses if you are uncertain about the subject.
Line 71: low -> shallow
Line 84 & 95: make the name of Genbank consistent
Line 107: which version of GetOrganelle. Please specify versions for all software.
Line 108: please add "using default parameters and databases" if you were using, otherwise specify them.
Line 108: Complete plastomes were viewed -> The assembly graph of the generated complete plastome was verified
Line 116-120: Rephrase the entire paragraph. It is about usage codon bias.
Line 130 & 131: Adobe Illustrator? Remember to specify the versions.
Line 144: Gblocks removes ambiguously-aligned positions, which is different from the most variable positions.
Line 187: Heatmap analysis? Please do not refer to an analysis using the visualization tool unless it is only for visualization.
Line 240: 43 bp -> 43 bp to 45 bp?
Line 282: for -> within
Line 283: few -> a few
Line 299: I do not think the limited evidence in this study can support that S. honania and S. meiliensisi are likely to be conspecific.
Line 305: Sequence length .. lead to this result?
Round 2
Reviewer 1 Report
Reply to reviewer #1
Generally, the manuscript lacks discussion. There are just a few sentences at the end of
each subtitle. This part should be most important and the current form is unacceptable.
I would recommend separation results from discussion in this particular case.
Reply: We found that not every part is suitable for in-depth discussion, so we put the results and
discussion together. If the two parts must be separated, we will revise them in next round.
Well, merging results and discussion didn’t solve the main problem of poorly developed the former. There’s a lot of papers to discuss every aspect of this work. One again, merging is to the issue, but it doesn’t justify the lack of well developed discussion.
The matrices for phylogenetics partitions shouldn’t be selecter a priori. The whole
alignment should be partitioned according to evolutionary models using software like
partitionfinder, and then one final tree should be calculated using MrBayes or RAxML
(or both). It can be easily done using pipelines like Phylosuite. Subtrees can be
calculated using f.e. CDS and IGS/intronic dataset. Moreover, the manuscript would
gain from wider selection of Salvia plastomes - there’s over 100 present in GenBank.
Reply: We used only tree inferred from complete plastid genome in the revised manuscript with
other phylograms moved into the supplement.
Yes, you did, but removing artificial partitions doesn't solve the problem. Have you partitioned using any dedicated software? No. Have you expanded your dataset with available plastomes? No.
Codon usage analysis should be evaluated using modern statistical software using more
advanced models like MILC. There are no details about therehold used during this work
and CU analysis confidence suffers from short, usually below 80 AA long sequences.
Reply: The RSCU values for all protein-coding sequences were computed using the program codon
W 1.4.4. In fact, we also ran the analysis with Mega, and the results were consistent.
Both software calculate only basic statistics without pointing out statistically important differences in CU bias. Therefore I recommended a more advanced approach - coRdon package could be used to calculate MILC and other stats.
Conclusions should be rewritten too, since half of this paragraph has no conclusive
character but rather summarizing the results.
Reply: revised
OK
Next improvement could be made on molecular species identification, which is not
mentioned at all. The plastid superbarcodes are gaining popularity and sequenced
plastomes should be analyzed using one the modern species delimitation approach (DN,
MDCs or one of the tree based methods).
Reply: we plan to conduct this analysis in further study with more taxa sampled.
In the current verison results are still only descriptive. I don’t think it’s worth publishing without expanding data analysis toward a pointed direction.
Lines 266-270: only 3 out 7 references are about barcoding. Comparative analysis is
not a barcoding approach. And trnH-psbA is unefficient barcode for Lamiaceae
(Krawczyk et al. 2014).
Reply: revised. Is “Krawczyk K, Szczecińska M, Sawicki J. Evaluation of 11 single‐locus and
seven multilocus DNA barcodes in L amium L.(Lamiaceae). Molecular ecology resources, 2014,
14(2): 272-285.” Krawczyk et al. 2014? In this study, trnH-psbA together matK are demonstrated
to be the most effective in the identification of Lamium (Lamiaceae).
As you notics “together with matK” dual locus barcodes are tested in different ways, and in the case of trnH-psbA, the single locus barcodes fall short in species delimitation of most taxonomic groups.
Tables 1 and 2 should be merged removing non-variable columns from Table 2.
Additionally the coverage statistics should be given within this table. I would also
recommend providing SRA accession for sequenced reads.
Reply: we added information of clean reads, reads used assembly and coverage to table 2. Due to
insufficient table space after adding data, the two tables were not merged. In addition, the SRA
accession number cannot be provided at present, because we will do further data analysis later.
I would not recommend publication of any paper which didn’t meet data availability criteria. Even more specialized journals (like Mitochondrial DNA Resources) require availability of SRA dataset.
An analysis presented in figure 5 should be normalized by the total length of each part.
It would be nice to visualize distribution of repeats in a consensus genome map and
check the dispersion of repeats at phylogenetic trees.
Reply: We believe that this is not the focus of this study, so we did not conduct further statistics on
this part, and we have not seen other literatures suggesting that this part is related to phylogeny
Estimating frequencies without normalization is a common issue in many published papers. It’s not worth following.
Figure 6 should be removed, since it is uninformative - all plastomes are identical,
without any changes at boundaries.
Reply: The rps19 gene spanned the LSC/IRb boundary in all species with a length varying slightly
from 234 bp to 236 bp in LSC and 43 bp to 45 bp in IRb... So we kept this figure in revised
manuscript.
If you considered 2bp differences as worth mentioning and filling the whole page, I have nothing to add.
Figure 7 - please spit it in at least four parts, since it’s unreadable in present form
Reply: revised.
OK
Figure 8 - as the figure 6, it brings nothing to the table.
Reply: revised and moved to Appendix Figure S1 as support material.
OK
Author Response
Dear colleague, thanks very much for your efforts to our manuscript. We have revised manuscript. Please see the attachment.

Reviewer 2 Report
Dear authors, I appreciate your efforts in improving your manuscript. Thank you very much for the answers given to my comments. However, I found a few elements (minor flaws) which need your attention. Detailed comments to the new version of the manuscript are provided below in the detailed review report.
L.169
delete “and”
L.205-206
Redundancy: “...indicating that plastid genome in land plant the plastid genome is highly conserved”
L.220
replace “microsatellites” with “microsatellite”
L.297
I think that “among” will be better here than “within”
Table 2
I suggest a little improvement of two table headings:
- replace “Reads used assembly” with “ Reads used for genome assembly”
- replace “Protein-coding genes” with its abbreviation i.e. “PCG” and in the footnote explain the used abbreviation
Figure 8 caption
replace “Pi > 005” with „Pi > 0.005”
Author Response
Dear colleague, thanks very much for your efforts to our manuscript. We have revised manuscript according your suggestions. Please see the attachment.
